# Gait Analysis to Monitor Fracture Healing of the Lower Leg

**DOI:** 10.3390/bioengineering10020255

**Published:** 2023-02-15

**Authors:** Elke Warmerdam, Marcel Orth, Tim Pohlemann, Bergita Ganse

**Affiliations:** 1Werner Siemens-Endowed Chair for Innovative Implant Development (Fracture Healing), Clinics and Institutes of Surgery, Saarland University, 66421 Homburg, Germany; 2Department of Trauma, Hand and Reconstructive Surgery, Clinics and Institutes of Surgery, Saarland University, 66421 Homburg, Germany

**Keywords:** implant, non-union, malunion, motion capture, movement analysis, rehabilitation, tibial fracture, trauma, wearables

## Abstract

Fracture healing is typically monitored by infrequent radiographs. Radiographs come at the cost of radiation exposure and reflect fracture healing with a time lag due to delayed fracture mineralization following increases in stiffness. Since union problems frequently occur after fractures, better and timelier methods to monitor the healing process are required. In this review, we provide an overview of the changes in gait parameters following lower leg fractures to investigate whether gait analysis can be used to monitor fracture healing. Studies assessing gait after lower leg fractures that were treated either surgically or conservatively were included. Spatiotemporal gait parameters, kinematics, kinetics, and pedography showed improvements in the gait pattern throughout the healing process of lower leg fractures. Especially gait speed and asymmetry measures have a high potential to monitor fracture healing. Pedographic measurements showed differences in gait between patients with and without union. No literature was available for other gait measures, but it is expected that further parameters reflect progress in bone healing. In conclusion, gait analysis seems to be a valuable tool for monitoring the healing process and predicting the occurrence of non-union of lower leg fractures.

## 1. Introduction

Lower leg fractures include fractures of the tibia or fibula or of both bones combined. The incidence of tibial fractures in Sweden was 52 per 100,000 per year, with proximal tibial fractures (Figure 1) being the most common tibial fractures, followed by tibial shaft and distal tibial fractures, respectively 26.9, 15.7, and 9.1 per 100,000 per year [1]. The incidence of malleolar fractures in Sweden was 152 per 100,000 per year [2]. In some cases, non-displaced fractures are treated conservatively by immobilization using a cast or brace [3]. However, the majority of fractures require surgical treatment, either via external fixation, intramedullary nailing, or via open reduction and internal fixation by screws and/or plates. The mode of surgical fixation depends on the fracture location, fracture type, and soft tissue status (Figure 1). In some cases, e.g., after intramedullary nailing in a tibial shaft fracture, full weight bearing is possible immediately after surgery. In most cases, patients are only allowed partial weight bearing for six weeks or longer, although the benefits of partial weight bearing are currently being re-evaluated [4,5].

Delayed union and non-union are frequent complications in the treatment of long bone fractures and are associated with morbidity, repeated hospitalization, disability, significant functional limitations, and high costs [6,7,8,9]. Definitions of when to classify a non-healing fracture as non-union vary, but a healing period of at least six months after the fracture and more than three months without biological progression seem to be the recent consensus [10,11,12]. Non-union occurs in about 5% to 14% of tibial fractures [13,14] and is classified as either hypertrophic (abundant callus formation) or atrophic (no callus formation) [15]. Known contributing factors are multiple and open fractures, as well as smoking, alcoholism, diabetes, high body mass index, and male sex [13]. Biomechanical and biological factors both play crucial roles in fracture healing and include the fracture gap size, implant stiffness, and the extent of soft tissue damage [16]. Delays in bone healing may currently either be addressed by non-invasive interventions such as pulsed ultrasound, shockwaves and electromagnetic fields, or by revision surgery [11,17].

In clinical practice, the progress of fracture healing is typically monitored by infrequent radiographs that come at the cost of radiation exposure and reflect fracture healing with a time lag, as fracture mineralization occurs later than the increases in stiffness [18]. The inability to monitor the current healing progress in a timelier manner often results in treatment delays. In experimental studies, changes in stiffness or displacement were monitored continuously at the fracture site through sensors attached to the implants. This approach is, however, not yet established for daily clinical routine today and requires special and more expensive implants [19,20]. In addition, it would be desirable to predict fracture healing problems as early as possible to intervene sooner. It thus seems beneficial if further non-invasive diagnostic options could be made available to monitor the progress of fracture healing. Advanced gait analysis is a candidate that might enable for such predictions, potentially by combining several relevant gait parameters and by using regression models or machine learning.

During routine clinical visits, the gait pattern and ability to walk are usually visually inspected by the healthcare professional. Objective gait measures validated to predict the occurrence of non-union have, however, not yet been identified. It is highly desirable to explore such parameters and the possibility to monitor fracture healing via objective gait analysis, i.e., by optical motion capture, wearable sensors, or ground reaction force measurements.

In this narrative review, we provided an overview of what is known about characteristic gait changes throughout the fracture healing process and which gait parameters might be candidates to predict malunion or non-union of the lower leg early on. We included studies with both, surgical and non-surgical treatment. We also present suggestions for which gait aspects should be explored in future studies as potential parameters to predict fracture healing problems.

## 2. Materials and Methods

Pubmed was searched for studies that analyzed gait-related measures after lower leg fractures. A combination of search terms was used to find relevant studies for this review. The search terms at least included a part about gait (“gait” OR “walking”), fractures (“fracture”), and the lower leg (“lower leg” OR “tibia” OR “fibula” OR “ankle”). Additional search terms indicating a specific measurement method or gait-related parameters were occasionally added.

Studies with conservative or surgical treatment were included. Studies about stress fractures were excluded. Studies in languages other than English, German, or Dutch were also excluded. The references of the studies that were included were manually checked for other studies on this topic. In total, thirty studies were found that measured gait-related parameters at least once after a lower leg fracture (Table 1).

## 3. Changes in Gait throughout the Healing Process after Lower Leg Fractures

### 3.1. Spatiotemporal Gait Parameters

Spatiotemporal gait parameters are frequently measured to determine gait deviations. There are several systems available that can be used to extract spatiotemporal gait parameters, such as 3D optical motion capture systems [35,43], electronic walkways (pressure-sensitive mats) [8,44], and inertial measurement units (IMUs) [27]. The spatiotemporal gait parameters can be divided in different gait domains, i.e., pace, rhythm, variability, and asymmetry [49,50]. Spatiotemporal gait parameters that have already been assessed in the context of lower leg fractures are described below per gait domain. In all the described studies, the patients walked at their self-selected speed on an over-ground walkway.

#### 3.1.1. Pace

The pace-related gait domain consists of the parameters gait speed and step length. In the context of lower leg fractures, we found four studies that measured gait speed and three of these studies also measured step length throughout the healing process after proximal tibial, tibial shaft and malleolar fractures (Figure 2) [35,37,45,46]. These studies showed significant increases in gait speed and step length of the injured side over time. Twelve months after surgery, one study with tibial plateau fractures and another study with tibial shaft fractures did not find significant differences in gait speed anymore compared to healthy controls [8,37]. Multiple other studies that presented cross-sectional data ranging from about two months to five years after surgery or injury from patients with all types of lower leg fractures showed that gait speed and step length remained significantly shorter in patients compared to controls [7,29,30,31,35,39,42,43,44].

#### 3.1.2. Rhythm

Gait parameters from the rhythm domain that were reported by studies analyzing gait after a lower leg fracture were single-limb support time, cadence, swing time, stance time, and step time. Only the results for the injured side will be described below. Single-limb support time was the most frequently reported rhythm-related parameter. In some studies, it was reported in percentage of the gait cycle [7,44,45], while in others, it was presented in seconds [8,37,38,42]. From 6 to 12 months after surgery, cadence and step time increased in proximal tibial and tibial shaft fractures [25,51]. The single-limb support time increased between 6 and 12 weeks after surgery in patients with malleolar fractures but remained significantly shorter compared to healthy controls [45]. The single-limb support time did not change between 6 and 12 months after surgery for tibial shaft fractures [37]. The few studies that compared rhythm-related parameters with healthy controls showed that the single-limb support time between six weeks and three months after surgery and the cadence at four months and three years after surgery were both lower in patients after proximal tibial and malleolar fractures [7,31,44,45]. The stance time was shorter in the injured leg compared to the non-injured leg, but there was no significant difference between the injured leg and controls six to eight weeks after surgery for proximal tibial fractures [29]. Another study also found no significant differences between the stance time of the injured leg and that of healthy controls about three months after malleolar fractures [45]. Several other studies only reported rhythm-related parameters measured at one time point and did not compare their results with healthy controls, which makes it difficult to draw conclusions from these data with regard to the present questions [8,38,42,48].

#### 3.1.3. Variability

The stance time and swing time variability in patients after proximal tibial and tibial shaft fractures were reported only by one and two studies, respectively. The swing time variability showed 8% asymmetry between the injured and non-injured leg 12 months after frame removal following ring fixation of proximal tibial fractures in 23 patients [8]. The stance time variability was increased in the injured leg following a tibial shaft fracture in 49 patients, and it decreased significantly from 6 to 12 months after surgery [37]. It remained slightly higher in the injured leg compared to the non-injured leg five years after surgery (significance not tested) in another study with 29 patients with tibial shaft fractures [38].

#### 3.1.4. Asymmetry

Four studies reported one or more asymmetry-related parameters after proximal tibial and tibial shaft fractures [8,35,37,38]. Step length asymmetry is the percentage difference in step length between the injured and non-injured sides. Step length asymmetry decreased throughout the first months after surgery for tibial shaft fractures in 23 patients from 18.2% at two months to 5.5% at six months but was still higher six months after the surgery compared to healthy controls [35]. From the 6th to the 12th month after surgery, step length asymmetry, single support time asymmetry, and swing time asymmetry decreased significantly in 49 patients with tibial shaft fractures [37]. One study reported symmetry values based on the trunk acceleration signal characteristics in anterior–posterior and vertical directions in 10 patients with malleolar fractures [31]. Only the symmetry in the vertical acceleration signal was lower compared to healthy controls about four months after injury.

#### 3.1.5. Spatiotemporal Gait Parameters Are Associated with Gait Speed

It is known that gait speed is associated with gait-related parameters in healthy adults [52,53]. Gait-related parameters improve with an increase in gait speed. We have pooled the data from the studies that measured gait speed and step length of the injured side that was described in the preceding paragraphs. We used the average values of these lower leg fracture studies and weighted them by the number of patients in the study. A weighted correlation showed a significant linear association between gait speed and step length (R^2^ = 0.85, *p* = 0.001, Figure 3), as expected based on the literature [54,55]. For the rhythm-related and variability-related gait parameters, there are currently not enough lower leg fracture studies available that could be used to pool the data to analyze a potential correlation. However, from existing literature on healthy adults, we know that the rhythm-related parameters change with gait speed. With higher speed, the cadence increased, and the temporal parameters decreased [52,53]. The spatiotemporal variability-related parameters slightly decrease with increasing walking speed [56] and increase with a decrease in walking speed [57]. The asymmetry-related gait parameters do not seem to be influenced by gait speed [58].

### 3.2. Kinematics

Kinematics are regularly assessed to obtain joint angles during a specific phase of the gait cycle or to obtain the range of motion (ROM). To measure kinematics, several technical solutions are available. To date, the gold standard is 3D optical motion capture systems that assess the exact position of markers located on the body and their position changes during movement in the 3D space [23,35,47]. There are also systems available that can be more easily used outside of the lab, such as IMUs, [27] and markerless motion capture based on video data [59]. IMUs are wearable sensors that are usually fixed to one or multiple body segments and measure acceleration and angular velocity. By placing IMUs on consecutive body segments, the joint angle between these two segments can be calculated. Markerless motion capture systems often use multiple synchronized cameras to detect the pose of one or multiple individuals based on machine learning models [59].

We found two studies that analyzed joint kinematics during walking more than once during the healing process after tibial fractures [23,35]. In one study with proximal tibial fractures, 18 patients were measured six times in the two years after the fracture [23]. This study showed that the hip functions had already returned to a normal pattern after six weeks, whereas for the knee and ankle, it took 26 weeks to function close to a normal pattern. Even after two years, the maximum knee flexion and the plantarflexion were still smaller in the injured leg compared to healthy controls. In the other study, 23 patients with tibial shaft fractures were measured three times within six months after the fracture [35]. This study showed that the knee flexion of the injured leg during the swing phase already returned close to normal in the first three months. Between three and six months post-surgery, the ankle plantarflexion during pre-swing and hip extension during stance improved but remained smaller compared to controls. Knee flexion during loading response and ankle dorsiflexion during the stance phase did not significantly increase throughout the 6 months and remained significantly smaller compared to controls.

Several studies that measured the kinematics after lower leg fractures showed that not all joint kinematics returned to similar values as controls [27,29,30,39,43,47]. This could partly be due to the lower walking speed of patients compared to controls, as gait speed is known to influence joint kinematics [52,53]. However, two studies compared the kinematics of patients at normal speed with controls walking at slow walking speeds, measuring at rather similar gait speeds, and still found significant differences between the groups [30,43]. Moreover, the location of the fracture might influence which joint kinematics return (faster) to control-like values and which remain altered. In patients with a proximal tibial fracture, the maximum knee flexion angle during the swing phase remained lower compared to controls after two years [23], whereas in patients with a tibial shaft fracture, the knee flexion during the swing phase returned close to normal in the first three months after surgery [35].

### 3.3. Kinetics

The forces and moments acting on the body during walking are often measured with force plates and optical motion capture systems. Changes in ground reaction forces, joint moments, and the generated power during walking throughout the rehabilitation process after lower leg fractures are described below.

#### 3.3.1. Ground Reaction Forces

The vertical ground reaction force (vGRF), measured by force plates or pressure sensors during walking, is characterized by a double-peak pattern (two maxima with a minimum in between). The first peak occurs during loading and the second peak during push-off [32,34,35]. The height of the peaks is normally presented as a percentage of body weight. The two peaks did not significantly increase between the second and third month after surgery, but they did significantly increase from the second and third month to the sixth month after surgery in patients with tibial shaft fractures [35]. The two maxima were, however, still significantly lower compared to healthy controls six months after surgery. This could, at least partially, be because of the lower gait speed of the patients since vGRF is smaller in lower gait speeds [52]. Three to six months after the removal of a circular Ilizarov frame, a type of external fixator that can be used to treat fractures, there were no differences in peak forces anymore between the injured and non-injured leg in patients with proximal tibial fractures [32]. Another study used the vGRF to determine the changes in weight bearing of patients with tibial fractures over time and found a moderate correlation between weight bearing with fracture stiffness [34].

#### 3.3.2. Joint Moments

Joint moments are calculated based on the kinematic data and the GRF data during walking. Two studies that measured joint moments while walking multiple times after the surgery in patients with tibial fractures showed significant increases in hip, knee, and ankle moments during the stance and loading response phases within the first six months after surgery [35,46]. Between 6 and 12 months after surgery, the knee joint reaction forces did not significantly increase further [46]. Six months after the surgery, joint moments were still significantly lower in patients compared to controls [35]. Another study that measured 12 months after surgery in patients with malleolar fractures showed significantly lower plantar flexion moments in patients compared to controls but did not find differences in dorsal flexion moments between the groups [39].

#### 3.3.3. Generated Power

Two studies looked at how much power was generated in the knee and ankle joints during walking [29,35]. The generated power is calculated as a product of the moment and angular velocity of the joint. The generated power increased significantly between two and six months after surgery in patients with proximal tibial fractures but remained significantly lower compared to controls at six months [35]. About three months after a proximal tibial fracture, the generated power was lower in the injured leg compared to the non-injured leg and compared to controls [29].

### 3.4. Pedography

Pedography captures the plantar pressure of the foot with a pressure plate or with pressure-measuring insoles. Studies have shown that at least one year after a lower leg fracture, the plantar pressure has moved more laterally compared to the non-injured side [21,22,28,33,40]. Especially under the fourth metatarsal, the pressure increased compared to the non-injured side in distal tibial fractures, and there was less loading in the heel and first metatarsal region [28,33]. Similar results were found for the force-time integral, which was higher in the fourth and fifth metatarsal regions and lower in the heel and first metatarsal regions [28,33]. The pedographic results seem to correlate with clinical scores in distal tibial fractures, such as the American Orthopeadic Foot and Ankle Society score, visual analog scale, Phillips scores, Ovadia-Beals score, Teeny-Wiss score and Takura score [21,33].

Two studies used pressure insoles to measure patients during daily living [24,36]. In one study where the insole was worn in a walking boot, a posterior shift of the center of pressure was found in seven patients in the weeks following surgery for tibial shaft or malleolar fractures [36]. The other study that analyzed 10 patients after malleolar fracture identified significant correlations between weight bearing during daily living and the visual analog pain scale, the Olerud–Molander score, and the American Orthopeadic Foot and Ankle Society score [24].

### 3.5. Muscle Activity and Mass

Electrical activity in the muscle can be measured with electromyography. Little is known about muscle activity after lower leg fractures. A cross-sectional study showed that muscle activity patterns did not significantly differ between the injured and non-injured leg in 17 patients that were measured once between 9 months and 14 years after free-flap reconstructions because of open tibial fractures [40]. They found, however, high variability in muscle activity patterns between patients.

Muscle cross-sectional area, as well as muscle volume of the lower leg, decreased during the immobilization phase with a cast after ankle fractures, measured in 18 patients with longitudinal magnetic resonance imaging [41]. The cross-sectional area of the tibialis anterior, gastrocnemius medialis, and gastrocnemius lateralis muscles stopped decreasing after about 30 days in a cast, whereas the soleus muscle was still decreasing at the last measurement point at 43 days in a cast. The total muscle volume in the injured leg was reduced by 16% due to immobilization and in the non-injured leg by 7%. Another study showed that after the removal of the cast following a malleolar fracture, the muscle strength was significantly lower in the injured leg compared to the non-injured leg [26]. Following a rehabilitation program, there were no significant differences anymore between the injured and non-injured leg at three and six months after cast removal.

## 4. Predicting Non-Union Based on Gait

About 5% to 14% of patients experience delayed union or non-union following lower leg fracture [13,14]. Multiple studies with large sample sizes will be needed to be able to determine which gait factors can predict union problems. So far, only a few studies have reported gait parameters in patients with union problems after fractures of the lower leg. Patients with delayed union were only able to bear 40% of their weight at 20 weeks post-surgery, measured with pedography, whereas patients with union were able to bear their full weight at that time [34]. Another study that measured pedography data with insoles in seven patients with tibial shaft or malleolar fractures for several weeks showed that one patient with non-union shifted the center of pressure (COP) anteriorly and that it remained near the forefoot over time, whereas the patients with union moved their COP posteriorly toward the heel [36]. Another study reported about patients with unsuccessful results (not solely non-union) and showed decreased loading of the lateral forefoot, whereas patients with successful results showed an increased loading of the lateral forefoot compared to the non-injured side [22]. From the described studies in this review, only studies with pedography assessed differences in gait-related parameters in non-union patients. The results from these studies are promising; however, a larger sample size is required to really draw conclusions from the data. Moreover, other gait measures than pedography should be explored regarding their potential to predict non-union early. We expect that also, in the spatiotemporal, kinematic, and kinetic gait measures, there will be parameters that have the potential to predict non-union. A combination of multiple gait-related parameters that change with fracture healing will provide a more accurate prediction of the healing process.

## 5. Discussion

In this review, we provided an overview of how gait changes throughout the healing process of lower leg fractures in patients with union. We also explored whether gait can be used to predict union problems in a timelier manner.

Improvements were found in all gait-related parameters that were measured longitudinally after a lower leg fracture with normal healing. Spatial parameters, joint kinematics, kinetics, and weight bearing increased, and temporal parameters decreased in the months following a lower leg fracture. However, not all parameters returned to normal values within several months after the injury. Gait speed is an important parameter to monitor since it increases throughout the healing process, and many gait-related parameters improve with increasing gait speed. Only very few studies described gait-related parameters in patients with non-union fractures, but pedography showed clear differences between patients with union and patients with non-union.

Gait analysis to monitor fracture healing could reduce the number of radiographs that are currently made. Radiographs will still be required to diagnose the fracture, check the position of the implant, analyze fracture reduction, and make sure via a final radiograph that the fracture has healed properly. The number of radiographs during the healing phase can be reduced when gait analysis is used to monitor fracture healing, and the gait pattern improves as expected. The final radiographs to make sure the fracture healed properly could, for example, be made when the gait asymmetry has returned to normal values. This has the potential to reduce the number of clinical visits after a fracture.

There was a large variance between the studies in terms of the types of fractures, type of implants, partial weight bearing or immobilization, the time after injury that gait analysis was performed, the equipment used for the gait analysis, and the parameters extracted. Nonetheless, multiple types of gait-related parameters improved during the first year(s) after a lower leg fracture. Therefore, gait analysis might be a suitable tool to monitor the healing process.

Different types of lower leg fractures were included in this review. Based on the limited available data, it is difficult to say whether the gait pattern improves in a similar fashion or if there are differences among fracture types. Several studies compared different ankle fracture types with each other and found significant differences in gait parameters [27,30]. The spatiotemporal gait parameters were significantly better in patients with a unimalleolar fracture compared to bimalleolar and trimalleolar fractures [27]. Ankle fracture severity correlated with the ROM of the ankle during walking [30]. However, these studies have not analyzed how the gait pattern changes throughout the healing process by, i.e., comparing the longitudinal gait parameters in different fracture types.

Besides the type of fracture, also the type of implant can have an effect on the gait parameters, especially when comparing a nail versus a plate. Patients with a plate often need to adhere to restricted weight-bearing instructions throughout the first few weeks after the surgery. This will affect the gait pattern differently compared to patients with a nail, who are often allowed to fully weight-bear shortly after the surgery.

To be able to interpret gait changes, it is necessary to know the gait alterations typical for specific patient cohorts. It is known that several factors, such as age, body weight, and pain, influence the gait pattern. With increasing age, gait becomes more cautious, and reductions of the preferred gait speed, cadence, and step length can be observed, while gait variability and asymmetry measures remain stable over time [60,61]. In people with obesity, the spatiotemporal gait parameters, joint kinematics, and pedography show significant differences compared with normal-weight-matched control groups [62,63]. Pain is also known to influence gait. Plantar heel pain causes reductions and changes in the GRF-based gait parameters [64]. Chronic joint pain is associated with poor gait performance, quantified by several spatiotemporal gait parameters [65]. Therefore, these kinds of factors should be taken into account when interpreting the gait data.

In the studies analyzed in this review, the average of all study age averages pooled is 44 years. However, most studies also provided the range of the age of the patients. The average number of years between the youngest and oldest participants is 48 years. This indicates that in most studies, the age range was rather large, which could have potentially led to differences within studies and also between studies.

Another important factor that should be taken into account in gait analysis is gait speed. Several gait-related parameters are known to change with gait speed [52,53]. It is currently unknown whether the improvements in gait after a lower leg fracture are solely due to an increase in gait speed or whether the gait pattern also improves besides the improvements due to an increase in gait speed. Since gait speed has a large effect on several gait-related parameters, it appears to be one of the most important parameters to track during the healing process.

Several gait-related parameters were still significantly different from controls months to multiple years after the injury. The current literature and data do not explain why these parameters do not return to control-like values, but possible reasons include persisting pain, damage and scarring of the soft tissues due to the injury and surgery, losses of sensation and proprioception, alterations of neuromuscular interaction, as well as losses in muscle strength and mass. Patients lose muscle strength and mass due to the immobilization and restrictions in weight bearing during the first weeks after the fracture [41]. Moreover, patients are, in general, likely to be less active because they are limited in the activities that they can perform. After immobilization or restricted weight bearing, the muscle mass and strength will increase again. Three months after the removal of the cast, there was no longer a difference in muscle strength between the injured and non-injured leg [40]. Similar results were found after 90 days of bed rest; the calf muscle cross-sectional area returned to baseline values after about 100 days [66]. However, in many patients with fractures, muscle strength never returns to pre-injury values [67].

Multiple studies showed that several patients were not pain-free one or multiple years after a lower leg fracture [28,38,39]. An association between the pain level in the foot and the gait performance measured with the Edinburgh visual gait score was shown based on normal video recordings in patients with malunited tibia fractures treated with external fixation [68]. It is also known that in other medical conditions with pain, such as plantar heel pain and chronic joint pain, the gait pattern is different from healthy controls [64,65]. Therefore, pain after the injury certainly contributes to gait changes.

Several other factors that are known to influence the gait pattern, in general, are age, body weight, physical function, and cognition [60,63,69,70]. Most of these factors will not change substantially during the healing process of a fracture and thus only have a small influence on the changes in the gait pattern. However, these demographics-induced differences in gait patterns between patients make it hard to compare fracture healing between patients. We, therefore, recommend looking at changes in the gait pattern within patients throughout the healing process. Moreover, especially in people at high age, with low physical function and comorbidities, these factors may influence the healing process and might need to be taken into account when monitoring the course of fracture healing based on the gait pattern. In this population, a period of inactivity can further reduce their function, and a large part of these patients might not return to pre-injury levels of mobility anymore, as was seen after hip fractures [71]. A high body weight will not only affect the gait pattern but might also have an effect on fracture healing. The healing process after a fracture took longer in obese mice compared to non-obese mice [72].

### 5.1. Predicting Non-Union Based on Gait

Several studies showed that gait analysis has the potential to detect healing problems based on gait-related parameters. The plantar pressure distribution and weight bearing were different in patients with healing problems [22,24,34,36]. Differences were found between healing and non-healing patients in pressure distribution already about 2 months after surgery for tibia shaft or malleolar fractures [36]. This is sooner than radiographs allow to determine healing problems. This shows that gait-related parameters have the potential to detect union problems early on. There are still many gait parameters that have not yet been analyzed in relation to bone healing problems. Longitudinal studies with a large sample size that include multiple patients with healing problems are required to analyze the relation between gait-related parameters and the occurrence of healing problems.

We expect that in patients without healing problems, gait-related parameters will improve over time and that there will be only minor improvements seen in patients with healing problems. This hypothetical relationship between time and healing in patients with and without healing problems is depicted in Figure 4. This relationship can be influenced by, e.g., type and severity of the fracture, soft tissue damage, and type of implant. Moreover, factors such as age, body weight, cognition, comorbidities, and fear of falling might determine which values the gait-related parameters will return to in patients without healing problems.

### 5.2. Suggestions for Future Research Analyzing Gait after Lower Leg Fractures

Gait analysis has the potential to monitor the fracture healing process. However, more data about changes in gait-related parameters throughout the healing process are required to determine how the gait changes in patients with union and non-union. Most of the described studies only measured the gait pattern once (Figure 5); however, to monitor the healing process, multiple measurements are required. We suggest analyzing the gait pattern multiple times within the first year. However, to analyze when and if the gait pattern fully returns to normal values, measurements might even need to take place beyond one year after the injury.

After a lower limb fracture, a large part of the patients is asked to perform partial weight bearing for a certain period of time. The time and duration of no or partial weight bearing differed quite a bit between the studies that reported this information. It is known that it is difficult to adhere to partial weight-bearing instructions [5,73]. Nonetheless, to be able to look into the effects of partial weight-bearing strategies or immobilization after a lower leg fracture, it would be helpful if future studies could report this information.

Several studies compared the gait-related parameters of the injured leg with the non-injured leg [32,33,37,38]. However, it could be that the gait pattern of the non-injured leg has also changed because of changes in walking speed or compensation and adaptation due to the injury in the other leg. Therefore, we recommend comparing the results to a control group with comparable characteristics (e.g., age, body weight) since they are known to influence the gait pattern [60,63,69,70]. In case the patients and the matched control group are not measured at the same gait speed, if possible, a correction for gait speed should be performed for the parameters that change with gait speed for a fair comparison between the groups. Gait speed should also be taken into account when comparing longitudinal measurements within patients.

Asymmetry parameters might be a suitable measure to quantify fracture healing. The asymmetry is high shortly after the fracture and decreases throughout the healing process. An advantage of asymmetry measures is that they are not influenced by gait speed and age [58,74]. Therefore, we recommend measuring the asymmetry of gait-related parameters throughout the healing process.

A very promising approach is to analyze signal characteristics of the gait data, such as via accelerometers. An accelerometer on the trunk was used to look at movement intensity by calculating the root mean square of the acceleration and studying symmetry and regularity based on the calculation of autocorrelation [31]. There are more signal characteristics, such as entropy, smoothness, Lyapunov exponent, and harmonic ratio, that could be explored to monitor changes in the gait pattern during fracture healing.

Most studies performed the data collection in the clinic and measured either only once or at multiple time points. With all the wearable devices on the market, it would also be possible to measure gait-related parameters continuously in the normal environment of the patient’s everyday life. Continuous data may provide additional and relevant information about fracture healing. We only found two studies that monitored gait continuously for several weeks after a lower leg fracture. These studies used pressure-sensitive insoles that were already described in the Section 3.4. To our knowledge, there have been no studies using other wearable devices for continuous monitoring of gait parameters after lower leg fractures. Tools that might be particularly suitable are accelerometers and/or gyroscope-based devices. These have already been used frequently in other patient groups for continuous measurements [75]. These devices can also be used to obtain simple quantitative gait or physical activity measures, such as the number of steps. Daily step count or physical activity is known to increase after lower leg fractures [76] and could potentially also be used to monitor fracture healing.

To implement gait analysis in routine clinical care, it needs to be clarified which gait parameters can differentiate between normal healing and delayed or non-union. Not all of the available equipment used for gait analysis is suitable for a quick assessment in clinical practice. Wearable sensors, such as pressure-sensing insoles and IMUs, are devices that can be used relatively fast and easily to quantify the gait pattern. Additionally, an appropriate software package that can instantly analyze the data and report on the gait quality and the healing process needs to be developed.

A limitation of this review is that there were few studies found that presented longitudinal patient data. The studies with a single measurement often measured at different time points after the injury. By combining those studies, it was still possible to see improvements in the gait pattern; however, more longitudinal studies would have been desirable.

## 6. Conclusions

Gait analysis can be used to determine changes in the gait pattern after lower leg fractures. In patients with union, the spatiotemporal parameters, joint kinematics, kinetics, and pedographic measures improved in the months following the fracture. Three studies with pedograpic measures were able to show differences in the changes in gait patterns during the healing process in patients with and without union. These differences in patients with healing problems were detected earlier than would have been possible with radiographs. Therefore, gait analysis can be used to monitor the healing process and to predict the occurrence of non-union of the lower leg. However, several steps still need to be taken before gait analysis can be implemented in routine clinical care to monitor fracture healing of the lower leg.

## Figures and Tables

**Figure 1 bioengineering-10-00255-f001:**
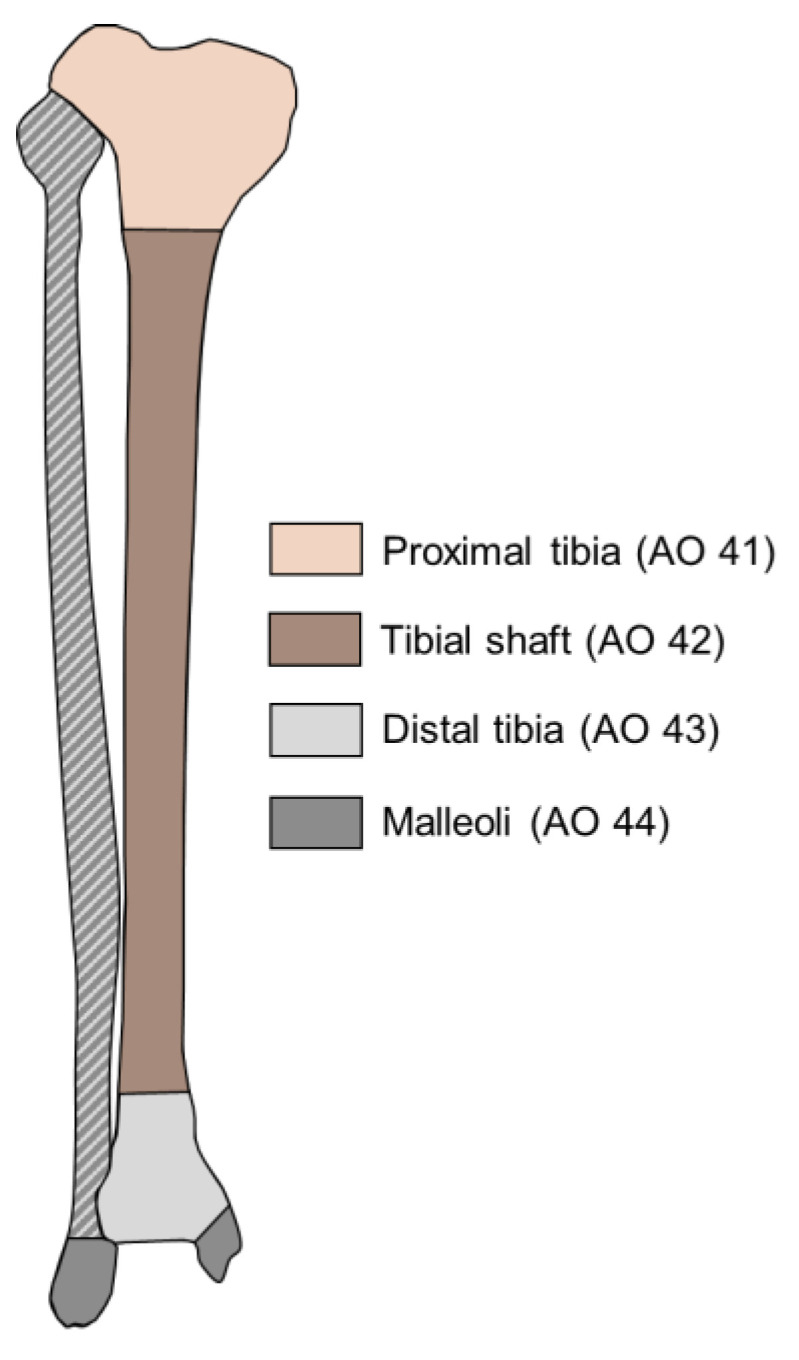
Fracture locations of the lower leg. The fibula can be involved in both distal tibial fractures as well as malleolar fractures. The distal tibia may also be involved in malleolar fractures. The AO classification numbers associated with the fracture locations are provided between brackets.

**Figure 2 bioengineering-10-00255-f002:**
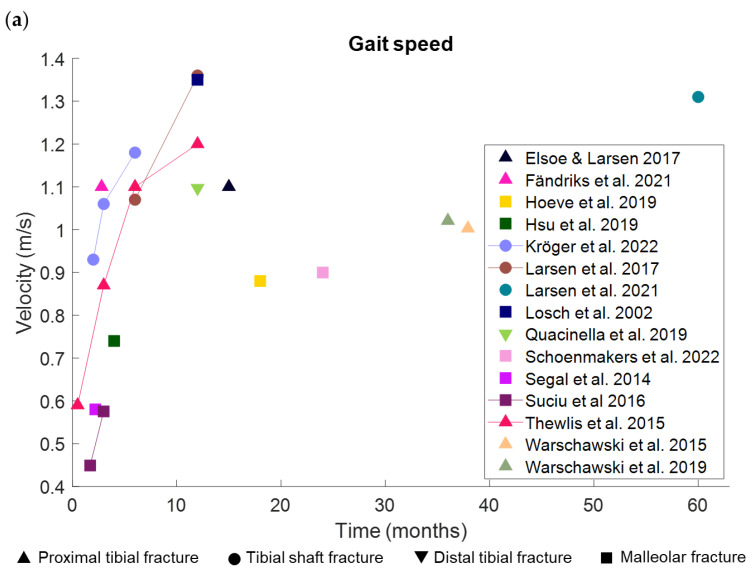
Overview of data from studies with different type of lower leg fractures [7,8,29,30,31,35,37,38,39,42,43,44,45,46,48]. (**a**) Gait speed after a lower leg fracture. (**b**). Step length after a lower leg fracture of the injured side. The type of lower leg fracture is indicated by the shape of the marker.

**Figure 3 bioengineering-10-00255-f003:**
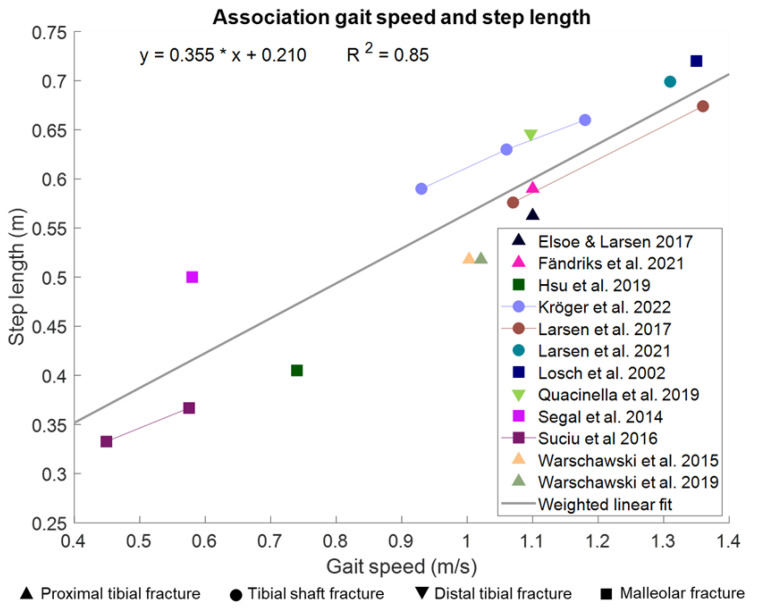
Linear relationship between gait speed and step length after a lower leg fracture [7,8,29,31,35,37,38,39,42,44,45,48]. Data pooled from studies with lower leg fractures, the correlation is weighted based on the number of participants per study.

**Figure 4 bioengineering-10-00255-f004:**
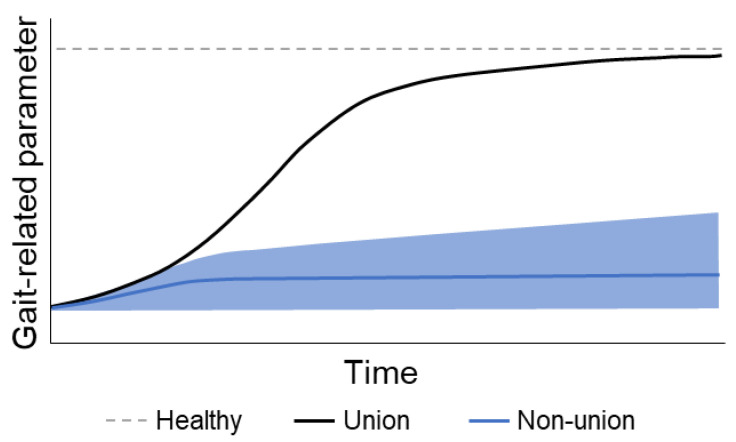
The expected changes in gait-related parameters after surgery in union and non-union of lower leg fractures. An increase indicates an improvement. The shaded blue indicates the expected range for non-union fractures.

**Figure 5 bioengineering-10-00255-f005:**
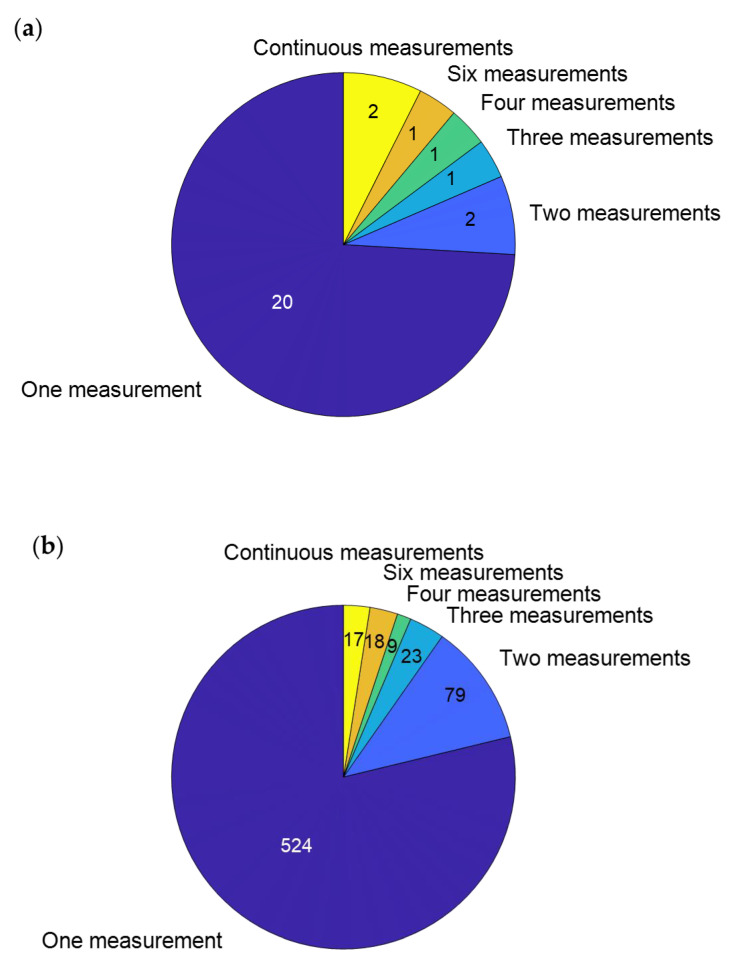
(**a**) Pie chart indicating how many studies performed a certain number of measurements. (**b**) Pie chart indicating how many patients from the studies combined were measured a certain number of times.

**Table 1 bioengineering-10-00255-t001:** Studies that measured gait-related parameters after lower leg fractures.

Authors	Fracture Type(s) Included	n	Age (Years)	Longitudinal Measurements	Time After Fracture/Surgery That Measurement Occurred	Measurement Device(s)	Type of Parameters Calculated
Agar et al., 2022 [21]	Intraarticular distal tibial fractures	62	43	No	24–58 months	Pressure plate	Pedographic measures
Becker et al., 1995 [22]	Malleolar fractures	40	24	No	18.5 months	Pressure plate	Pedographic measures
Bennet et al., 2021 [23]	Proximal tibial fractures	18	52	Yes	2 weeks, 6 weeks, 3 months, 6 months, 1 year, 2 years	Optical motion capture, force plate	Kinematics
Braun et al., 2016 [24]	Malleolar fractures	10	53	Yes	Continuously for 6 weeks	Pressure insoles	Pedographic measures
Deleanu et al., 2015 [25]	Proximal tibial fractures	25	39	No	Before hardware removal	Pressure plate, ultrasound-based motion capture	Spatiotemporal gait parameters
Ekinci et al., 2021 [26]	Malleolar fractures	24	41	Yes	At cast removal, 3 and 6 months after rehabilitation	Isokinetic dynamometer	Muscle strength
Elbaz et al., 2016 [27]	Malleolar fractures	24	49	No	<6 weeks from weight-bearing approval	Inertial measurement units	Kinematics
Elsoe et al., 2017 [8]	Proximal tibial fractures	23	54	No	12 months after ring fixator removal	Electronic walkway	Spatiotemporal gait parameters
Falzarano et al., 2018 [28]	Nonarticular distal tibial fractures	34	32	No	12 months	Pressure plate	Kinetics
Fändriks et al., 2021 [29]	Proximal tibial fractures	20	44	No	85 days	Optical motion capture	Spatiotemporal gait parameters, kinematics
Hoeve et al., 2019 [30]	Malleolar fractures	33	57	No	18 months	Optical motion capture, force plate	Kinematics
Hsu et al., 2019 [31]	Malleolar fractures	10	38	No	4 months	accelerometer	Spatiotemporal gait parameters
Iliopoulos et al., 2020 [32]	Proximal tibial fractures	16	49	No	3–6 months after frame removal	Force plate	Kinetics, spatiotemporal gait parameters
Jansen et al., 2013 [33]	Distal tibial fractures	41	48	No	50 months	Pressure plate	Pedographic measures
Joslin et al., 2008 [34]	Tibial shaft fractures	12	32	No	20 weeks	Force plate	Kinetics
Kröger et al., 2022 [35]	Tibial shaft fractures	23	39	Yes	2, 3, and 6 months	Optical motion capture, force plate	Kinematics, kinetics, spatiotemporal gait parameters
Lajevardi-Khosh et al., 2019 [36]	Tibial shaft fractures and malleolar fractures	7		Yes	Continuously for 2–12 weeks	Pressure insoles	Pedographic measures
Larsen et al., 2017 [37]	Tibial shaft fractures	49	43	Yes	6 months, 12 months	Electronic walkway	Spatiotemporal gait parameters
Larsen et al., 2021 [38]	Tibial shaft fractures	29	46	No	5 years	Electronic walkway	Spatiotemporal gait parameters
Losch et al., 2002 [39]	Malleolar fractures	20	43	No	1 year	Optical motion capture, force plate	Kinematics, kinetics, spatiotemporal gait parameters
Perttunen et al., 2000 [40]	Tibial fractures	17	51	No	9 months–14 years	Pressure insoles, electromyography	Pedographic measures, muscle activity
Psatha et al., 2012 [41]	Malleolar fractures	18	43	Yes	5, 8, 15, 29 and 43 days	Magnetic resonance imaging	Muscle volume
Quacinella et al., 2019 [42]	Distal tibial fractures	7	25	No	12 months	Optical motion capture, force plate	Spatiotemporal gait parameters
Schoenmakers et al., 2022 [43]	Malleolar fractures	26	58	No	24 months	Optical motion capture	Kinematics
Segal et al., 2014 [44]	Malleolar fractures	41	48	No	67 days	Electronic walkway	Spatiotemporal gait parameters
Suciu et al., 2016 [45]	Malleolar fractures	30	53	Yes	7 weeks, 12 weeks	Pressure plate	Spatiotemporal gait parameters
Thewlis et al., 2015 [46]	Proximal tibial fractures	9	69	Yes	2 weeks, 3 months, 6 months, 1 year	Optical motion capture, force plate	Kinetics
Wang et al., 2010 [47]	Malleolar fractures	18	39	No	1 year	Optical motion capture	Kinematics, spatiotemporal gait parameters
Warschawski et al., 2015 [7]	Proximal tibial fractures	22	46	No	3 years	Floor-based photocell system	Spatiotemporal gait parameters
Warschawski et al., 2019 [48]	Proximal tibial fractures	21	44	No	3 years	Floor-based photocell system	Spatiotemporal gait parameters

## Data Availability

Data can be extracted from the cited papers.

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
