# Peer review of "Gait Analysis to Monitor Fracture Healing of the Lower Leg"

_bioengineering, 2023, doi:10.3390/bioengineering10020255_

Round 1

Reviewer 1 Report

The topic of the study addresses an issue very current in the clinical practice. Non-union or malunion are frequent complications of lower leg fractures, whose early prediction and diagnosis could improve clinical outcome and reduce costs, disability, and hospitalization time.  

The abstract should be better reported: authors might better explain the topic of the issue, resuming in a more comprehensive way the purpose of the study.

The introduction is overall comprehensive and exhaustive but there are some parts that authors should be improve:

-        At lines 49-51, authors might provide an example for what they mean for “via implants”.

-        At line 52-54, if the authors consider it appropriate, definition and further overview of non-unions and delayed unions might be better explained mentioning also the following study       doi: 10.3390/bioengineering9100560 .

-        At line 28-29, authors underline that the most frequent tibial fractures are those involving the proximal segment. They might even add that the tibial pylon fractures account for about 1% of all lower limb fractures and 10% of all tibial fractures the article, mentioning the following article             doi: 10.3390/medicina58111641

Section 2 provides gait parameters and analysis which are carefully described by authors. They, in fact, compared all articles found in literature, about each parameters evaluated, and reported them in summary and detailed charts and sub-sections. All references are clearly reported. 

Section 3 is unclear and should be better improved. Do the authors think that only one parameter, and in the present case pedography, could be a good and early predictor of non-union or malunion?

Furthermore, even the discussion is fragmentary and unclear. The focus of the study is not well distinguished, authors can’t provide an overall view of the aim proposed. Comparison between parameters and articles reported appears confusing. Authors in fact should include a section about “material and methods” to describe the methodological approach used: talking about the database, and explaining which article they included from the literature, might be useful to make the study more comprehensive.

Author Response

We are thankful for the useful comments by the reviewer that have helped to strengthen our manuscript substantially. The changes we have made are highlighted in the manuscript. We hope the reviewers will now find our manuscript acceptable for publication. 

The topic of the study addresses an issue very current in the clinical practice. Non-union or malunion are frequent complications of lower leg fractures, whose early prediction and diagnosis could improve clinical outcome and reduce costs, disability, and hospitalization time.  

Comment 1: The abstract should be better reported: authors might better explain the topic of the issue, resuming in a more comprehensive way the purpose of the study.

Response 1: We thank the reviewer for the feedback on this review. We restructured the beginning of the abstract to make the purpose of this review clearer:

Fracture healing is typically monitored by infrequent radiographs. Radiographs come at the cost of radiation exposure and reflect fracture healing with a time lag due to delayed fracture mineralization following increases in stiffness. Since union problems occur frequently after fractures, better and timelier methods to monitor the healing process are required. In this review we provide an overview of the changes in gait parameters following lower leg fractures to investigate whether gait analysis can be used to monitor fracture healing.

The introduction is overall comprehensive and exhaustive but there are some parts that authors should be improve:

Comment 2:  At lines 49-51, authors might provide an example for what they mean for “via implants”.

Response 2: The changes in stiffness or displacement are measured by sensors that were fixated on the implant.  The following changes were made to the sentence to better explain what we mean:

In experimental studies, changes in stiffness or displacement were monitored continuously at the fracture site through sensors attached to the implants.

Comment 3:  At line 52-54, if the authors consider it appropriate, definition and further overview of non-unions and delayed unions might be better explained mentioning also the following study       doi: 10.3390/bioengineering9100560 .

Response 3: The classification of non-union was added to the introduction:

Non-union occurs in about 5 to 14% of tibial fractures [13,14] and is classified as either hypertrophic (abundant callus formation) or atrophic (no callus formation) [15].

Comment 4: At line 28-29, authors underline that the most frequent tibial fractures are those involving the proximal segment. They might even add that the tibial pylon fractures account for about 1% of all lower limb fractures and 10% of all tibial fractures the article, mentioning the following article             doi: 10.3390/medicina58111641

Response 4: We have added more detailed information about the incidence of the tibial fractures:

The incidence of tibial fractures in Sweden was 52 per 100,000 per year, with proximal tibial fractures (figure 1) being the most common tibial fractures followed by tibial shaft and distal tibial fractures, respectively 26.9, 15.7 and 9.1 per 100,000 per year [1]. The incidence of malleolar fractures in Sweden was 152 per 100,000 per year [2].

Section 2 provides gait parameters and analysis which are carefully described by authors. They, in fact, compared all articles found in literature, about each parameters evaluated, and reported them in summary and detailed charts and sub-sections. All references are clearly reported. 

Comment 5: Section 3 is unclear and should be better improved. Do the authors think that only one parameter, and in the present case pedography, could be a good and early predictor of non-union or malunion?

Response 5: Several changes to this section were made. We expect that a combination of gait parameters will be more accurate to predict the healing progression compared to one single parameter. This is not limited to pedographic measures. We expect that there are also several gait parameters from the other type of gait measures that are able to predict healing problems. There are, to the best of our knowledge, no available studies that looked into this.

Comment 6: Furthermore, even the discussion is fragmentary and unclear. The focus of the study is not well distinguished, authors can’t provide an overall view of the aim proposed. Comparison between parameters and articles reported appears confusing. Authors in fact should include a section about “material and methods” to describe the methodological approach used: talking about the database, and explaining which article they included from the literature, might be useful to make the study more comprehensive.

Response 6: The beginning of the discussion was changed to better clarify the aim of the study. A “material and methods” section was added to the review explaining the inclusion and exclusion criteria and the search strategy. Moreover, a table was added to this section with an overview of the included studies.

  1. Materials and Methods

Pubmed was searched for studies that analyzed gait-related measures after lower leg fractures. A combination of search terms was used to find relevant studies for this review. The search terms at least included a part about gait (“gait” OR “walking”), fractures (“fracture”) and the lower leg (“lower leg” OR “tibia” OR “fibula” OR “an-kle”. Additional search terms indicating a specific measurement method or gait-related parameters were occasionally added.

Studies with conservative or surgical treatment were included. Studies about stress fractures were excluded. Studies in languages other than English, German or Dutch were also excluded. The references of the studies that were included were manually checked for other studies on this topic. In total, thirty studies were found that measured gait-related parameters at least once after a lower leg fracture (table 1).

Reviewer 2 Report

I commend the authors for their research entitled “Gait analysis to monitor fracture healing of the lower leg”. The specific questions of the research are what is known about characteristic gait changes throughout the fracture healing process and which gait parameters might be candidates to predict malunion or non-union of the lower leg early on. The topic is discussed as a narrative review. Overall the manuscript is well written and covers existing methods in a well-structured and organised manner. Innovative and high-quality figures are supporting the main text.  The important references are included and addressed thoroughly. However, the Conclusions section is very general – may need to be a bit stronger by elaborating more on the findings and especially the limitations of gait analysis to monitor fracture healing in routine clinical practise, so in that way the conclusion of the manuscript will be stronger and more complete.     

Author Response

We are thankful for the useful comments by the reviewer that have helped to strengthen our manuscript substantially. The changes we have made are highlighted in the manuscript. We hope the reviewers will now find our manuscript acceptable for publication. 

I commend the authors for their research entitled “Gait analysis to monitor fracture healing of the lower leg”. The specific questions of the research are what is known about characteristic gait changes throughout the fracture healing process and which gait parameters might be candidates to predict malunion or non-union of the lower leg early on. The topic is discussed as a narrative review. Overall the manuscript is well written and covers existing methods in a well-structured and organised manner. Innovative and high-quality figures are supporting the main text.  The important references are included and addressed thoroughly. However, the Conclusions section is very general – may need to be a bit stronger by elaborating more on the findings and especially the limitations of gait analysis to monitor fracture healing in routine clinical practise, so in that way the conclusion of the manuscript will be stronger and more complete.    

Response: We thank the reviewer for the feedback on this review. Changes to the conclusion section were made to make it more specific. At the end of the discussion section a paragraph about the steps that need to be taken before gait analysis can be implemented in routine clinical care to monitor fracture healing:

To implement gait analysis in routine clinical care, it needs to be clarified which gait parameters can differentiate between normal healing and delayed or non-union. Not all of the available equipment used for gait analysis is suitable for a quick assessment in clinical practice. Wearable sensors, such as pressure-sensing insoles and IMUs are devices that can be used relatively fast and easy to quantify the gait pattern. Additionally, an appropriate software package that can instantly analyse the data and re-port on the gait quality and the healing process needs to be developed.

  1. Conclusions

Gait analysis can be used to determine changes in the gait pattern after lower leg fractures. In patients with union, the spatiotemporal parameters, joint kinematics, kinetics and pedographic measures improved in the months following the fracture. Three studies with pedograpic measures were able to show differences in the changes in gait pattern during the healing process in patients with and without union. These differences in patients with healing problems were detected earlier than would have been possible with radiographs. Therefore, gait analysis can be used to monitor the healing process and to predict the occurrence of non-union of the lower leg. However, several steps still need to be taken before gait analysis can be implemented in routine clinical care to monitor fracture healing of the lower leg.

Reviewer 3 Report

Nice review about Gait analysis to monitor fracture healing. 

Minor issues: 

1. In clinical practice, whatever the gait parameters change during the healing process, radio-graph still needed to perform for drawing a final diagnosis (fracture heals successfully). So what is threshold of the gait-related parameters come to when the patient could go for the final x-ray test?

2. A table List for studies included in the study is needed.

3. There are differences in gait between adults and chlidren, elderly, and this should be discussed in the review, emphasizing the influence of age on Gait and fracture.

4. when monitoring gait on patients, how to rule out the effect of internal fixation on gait changing? 

Author Response

We are thankful for the useful comments by the reviewer that have helped to strengthen our manuscript substantially. The changes we have made are highlighted in the manuscript. We hope the reviewers will now find our manuscript acceptable for publication. 

Nice review about Gait analysis to monitor fracture healing. 

Minor issues: 

  1. In clinical practice, whatever the gait parameters change during the healing process, radio-graph still needed to perform for drawing a final diagnosis (fracture heals successfully). So what is threshold of the gait-related parameters come to when the patient could go for the final x-ray test?

Response 1: We thank the reviewer for the feedback on this review. We would indeed still need several radiographs, however the number of radiographs throughout the healing process could be reduced when gait analysis is used to monitor fracture healing. The following text was added to the discussion:

Gait analysis to monitor fracture healing, could reduce the number of radiographs that are currently made. Radiographs will still be required to diagnose the fracture, to check the position of the implant, analyse fracture reduction, and to make sure via a final radiograph that the fracture healed properly. The number of radiographs during the healing phase can be reduced when gait analysis is used to monitor fracture heal-ing and the gait pattern improves as expected. The final radiographs to make sure the fracture healed properly could for example be made when the gait asymmetry has re-turned to normal values. This has the potential to reduce the number of clinical visits after a fracture.

  1. A table List for studies included in the study is needed.

Response 2: We added a table (table 1) with information from all the included studies.

  1. There are differences in gait between adults and chlidren, elderly, and this should be discussed in the review, emphasizing the influence of age on Gait and fracture.

Response 3: We added more detail to the paragraph that discusses the influence of demographics on the gait pattern and how this should be taken into account:

Several other factors that are known to influence the gait pattern in general are age, body weight, physical function and cognition [57,60,66,67]. Most of these factors will not change substantially during the healing process of a fracture and thus only have a small influence on the changes in the gait pattern. However, these demographics-induced differences in gait patterns between patients, make it hard to compare fracture healing between patients. We therefore, recommend to look at changes in the gait pattern within patients throughout the healing process. Moreover, especially in people at high age, with low physical function and comorbidities, these factors may influence the healing process and might need to be taken into account when monitoring the course of fracture healing based on the gait pattern. In this population, a period of inactivity can further reduce their function and a large part of these patients might not return to pre-injury levels of mobility anymore, as was seen after hip fractures [68].

  1. when monitoring gait on patients, how to rule out the effect of internal fixation on gait changing?

Response 4: The type of implant could indeed have an effect on the gait pattern. This was added to the discussion:

Besides the type of fracture, also the type of implant can have an effect on the gait parameters, especially when comparing a nail versus a plate. Patients with a plate of-ten need to adhere to restricted weight bearing instructions throughout the first few weeks after the surgery. This will affect the gait pattern differently compared to patients with a nail who are often allowed to fully weight-bear shortly after the surgery.

Reviewer 4 Report

This manuscript entitled “Gait analysis to monitor fracture healing of the lower leg” was primarily aimed to review alterations of gait parameters after lower limb fractures. The authors bring an interesting study, but there are still some problems that cannot up this article to a publishing level. In general, the language and grammar mistakes throughout the whole article should be carefully edited. Suggestions are listed in the specific comments below.

Specific comments:

1.     In the abstract part, it is recommended to provide the descriptions about the background before the aim., which means that you should put lines 12-15 at the first part of this abstract.

2.     In the introduction part, line 45-47, “Definitions of when to classify a non-healing fracture as non-union vary, but a healing period of six months after the fracture and more than three months without biological progression seem to be the recent consensus [9].”. Please cite more referenced here.

3.     In the introduction part, line 74, “In this narrative review, we provide an overview of what is known about…” please write this sentence in the past tense.

4.     In the 2.2. Kinematics part, line 184-185, “To measure kinematics, several technical solutions are available.” Can you be more specific about what you mean for several technical solutions?

5.     In the 2.3.1. Ground reaction forces part, line 231-233, “The two peaks do not significantly increase between the second and third month after surgery, but they do significantly increase from the second and third month to the sixth.” Please write it in the past tense.

6.     In the 2.5. Muscle activity and mass part, line 291, “…with longitudinal MRI”. Please give official explanation of the abbreviation “MRI”, since it first appeared in this article.

7.     In the 3. Predicting non-union based on gait part, line 310, “…with non-union shifted the COP anteriorly…”. Please also give official explanation of the abbreviation COP”.

8.     In the discussion part, it is recommended to provide a brief description of the aim in the first paragraph of the manuscript.

9.     In the discussion part, line 338-339, “Several studies compared different ankle fracture types with each other and found significant differences in gait parameters month after surgery in patients with tibial shaft fractures.” Please cite relevant papers here.

10.  In the discussion part, Line 446-447, “Several studies compared the gait-related parameters of the injured leg with the non-injured leg.”. It is recommended to cite references here.

11.  Some recently studies could be added in the discussion, such as:

The Effect of Application of Asymmetry Evaluation in Competitive Sports: A Systematic Review. Physical Activity and Health, 6(1), 257–272. DOI: http://doi.org/10.5334/paah.215

Explaining the differences of gait patterns between high and low-mileage runners with machine learning. Sci Rep. 2022, 12(1), 2981.

mpact of Gait Events Identification through Wearable Inertial Sensors on Clinical Gait Analysis of Children with Idiopathic Toe Walking. Micromachines 2023, 14, 277. https://doi.org/10.3390/mi14020277

12.  What are the limitations of this study? Please provide relevant description.

Please check the languag

Author Response

We are thankful for the useful comments by the reviewer that have helped to strengthen our manuscript substantially. The changes we have made are highlighted in the manuscript. We hope the reviewers will now find our manuscript acceptable for publication. 

This manuscript entitled “Gait analysis to monitor fracture healing of the lower leg” was primarily aimed to review alterations of gait parameters after lower limb fractures. The authors bring an interesting study, but there are still some problems that cannot up this article to a publishing level. In general, the language and grammar mistakes throughout the whole article should be carefully edited. Suggestions are listed in the specific comments below.

 Specific comments:

  1. In the abstract part, it is recommended to provide the descriptions about the background before the aim., which means that you should put lines 12-15 at the first part of this abstract.

Response 1: We thank the reviewer for the feedback on this review. We have changed the order and put the background before the aim to improve the abstract.

  1. In the introduction part, line 45-47, “Definitions of when to classify a non-healing fracture as non-union vary, but a healing period of six months after the fracture and more than three months without biological progression seem to be the recent consensus [9].”. Please cite more referenced here.

Response 2: The following two references were added to this sentence:

Großner, T.; Schmidmaier, G. Conservative Treatment Options for Non-Unions. Unfallchirurg 2020, 123, 705–710, doi:10.1007/s00113-020-00851-1.

 Padilla-Eguiluz, N.G.; Gómez-Barrena, E. Epidemiology of Long Bone Non-Unions in Spain. Injury 2021, 52, S3–S7, doi:10.1016/j.injury.2021.02.053.

  1. In the introduction part, line 74, “In this narrative review, we provide an overview of what is known about…” please write this sentence in the past tense.

Response 3: The sentence was rewritten in past tense.

In this narrative review, we provided an overview of what is known about characteristic gait changes throughout the fracture healing process and which gait parameters might be candidates to predict malunion or non-union of the lower leg early on.

  1. In the2.2. Kinematics part, line 184-185, “To measure kinematics, several technical solutions are available.” Can you be more specific about what you mean for several technical solutions?

Response 4: The rest of the paragraph is used to mention and explain the most common methods that were used to assess the kinematics.

  1. In the 2.3.1. Ground reaction forces part, line 231-233, “The two peaks do not significantly increase between the second and third month after surgery, but they do significantly increase from the second and third month to the sixth.” Please write it in the past tense.

Response 5: The sentence was rewritten in past tense.

The two peaks did not significantly increase between the second and third month after surgery, but they did significantly increase from the second and third month to the sixth month after surgery in patients with tibial shaft fractures [19].

  1. In the 2.5. Muscle activity and masspart, line 291, “…with longitudinal MRI”. Please give official explanation of the abbreviation “MRI”, since it first appeared in this article.

Response 6: “MRI” was changed to “magnetic resonance imaging” in the text.

  1. In the 3. Predicting non-union based on gait part, line 310, “…with non-union shifted the COP anteriorly…”. Please also give official explanation of the abbreviation COP”.

Response 7: Thank you for pointing out these mistakes. The following text was added at the first mentioning of the COP:

center of pressure (COP)

  1. In the discussion part, it is recommended to provide a brief description of the aim in the first paragraph of the manuscript.

Response 8: The following text was added at the beginning of the discussion:

In this review, we provided an overview of how gait changes throughout the healing process of lower leg fractures in patients with union. We also explored whether gait can be used to predict union problems in a timelier manner.

  1. In the discussion part, line 338-339, “Several studies compared different ankle fracture types with each other and found significant differences in gait parameters month after surgery in patients with tibial shaft fractures.” Please cite relevant papers here.

Response 9: The references have now been added to this sentence:

Several studies compared different ankle fracture types with each other and found significant differences in gait parameters [22,30].

  1. In the discussion part, Line 446-447, “Several studies compared the gait-related parameters of the injured leg with the non-injured leg.”. It is recommended to cite references here.

Response 10: Multiple references have been added:

Several studies compared the gait-related parameters of the injured leg with the non-injured leg [25,33,46,49].                                                                                                                                                     

  1. Some recently studies could be added in the discussion, such as:

The Effect of Application of Asymmetry Evaluation in Competitive Sports: A Systematic Review. Physical Activity and Health, 6(1), 257–272. DOI: http://doi.org/10.5334/paah.215

Explaining the differences of gait patterns between high and low-mileage runners with machine learning. Sci Rep. 2022, 12(1), 2981.

mpact of Gait Events Identification through Wearable Inertial Sensors on Clinical Gait Analysis of Children with Idiopathic Toe Walking. Micromachines 2023, 14, 277. https://doi.org/10.3390/mi14020277

Response 11: We prefer not the add the first two references, because movement patterns during sport are not that close related to movement patterns during fracture healing. The third paper is about algorithms to detect initial contact and final contact during toe walking. In our review we do not discuss how the gait parameters can be best calculated, because definitely for the wearable sensors there are many different algorithms available. Discussing algorithm performance or methods to extract the gait-related parameters does not fit the scope of this review in our opinion.

  1. What are the limitations of this study? Please provide relevant description.

Response 12: A limitation paragraph was added at the end of the discussion. The following text was added:

A limitation of this review is that there were few studies found that presented longitudinal patient data. The studies with a single measurement often measured at different timepoints after the injury. By combining those studies, it was still possible to see improvements in the gait pattern, however more longitudinal studies would have been desirable.

Round 2

Reviewer 1 Report

The authors accepted suggestions received and correctly reported them in the manuscript.

Reviewer 4 Report

All my questions have been well addressed, I recommend to accept now.